# Elevated plasma free thiols are associated with early and one-year graft function in renal transplant recipients

**Marie B. Nielsen** [1,2], **Bente Jespersen**[1,3], **Henrik Birn**[1,2], **Nicoline V. Krogstrup**[1,4], **Arno R. Bourgonje** [5], **Henri G. D. Leuvenink**[6], **Harry van Goor**[7] *, **Rikke Nørregaard**[3] *

**1** Department of Renal Medicine, Aarhus University Hospital, Aarhus N, Denmark, **2** Department of Biomedicine, Aarhus University, Aarhus C, Denmark, **3** Department of Clinical Medicine, Aarhus University, Aarhus N, Denmark, **4** Department of Nephrology, Copenhagen University Hospital Rigshospitalet, Copenhagen Ø, Denmark, **5** Department of Gastroenterology and Hepatology, University Medical Center Groningen and University of Groningen, Groningen, the Netherlands, **6** Department of Surgery, University Medical Center Groningen and University of Groningen, Groningen, the Netherlands, **7** Department of Pathology and Medical Biology, University Medical Center Groningen and University of Groningen, Groningen, the Netherlands

* h.van.goor@umcg.nl (HVG); rn@clin.au.dk (RN)

**Data Availability Statement:** Data cannot be made publicly available due to ethical concerns, as it is not possible to anonymise data sufficient for public

# Abstract

## Background

Reduced free thiols in plasma are indicative of oxidative stress, which is an important contributor to ischaemia-reperfusion injury (IRI) in kidney transplantation leading to kidney damage and possibly delayed graft function (DGF). In a post-hoc, exploratory analysis of the randomised controlled CONTEXT trial, we investigated whether higher (i.e. less oxidised) plasma levels of free thiols as a biomarker of reduced oxidative stress are associated with a better initial graft function or a higher GFR.

## Methods

Free thiol levels were measured in plasma at baseline, 30 and 90 minutes after reperfusion of the kidney as well as at Day 1, Day 5 and twelve months after kidney transplantation in 217 patients from the CONTEXT study. Free thiol levels were compared to the kidney graft function measured as the estimated time to a 50% reduction in plasma creatinine (tCr50), the risk of DGF and measured GFR (mGFR) at Day 5 and twelve months after transplantation.

## Results

Higher levels of free thiols at Day 1 and Day 5 are associated with higher mGFR at Day 5 (p<0.001, $r^2_{adj.}$ = 0.16; p<0.001, $r^2_{adj.}$ = 0.25), as well as with mGFR at twelve months (p<0.001, $r^2_{adj.}$ = 0.20; p<0.001, $r^2_{adj.}$ = 0.16). However, plasma levels of free thiols at 30 minutes and 90 minutes, but not Day 1, were significantly higher among patients experiencing DGF.

access. Data is available on request to the
CONTEXT Data Access Committee from Lotte
Serwin (lottserw@rm.dk).

**Funding:** The Danish Council for Independent
Research (NVK), the Danish Society of Nephrology
(NVK), the Lundbeck Foundation (NVK), the Novo
Nordic Foundation (NVK), Nyreforeningen (the
Danish kidney patient association) (NVK), A.P.
Møller og hustru Chastine Mc-Kinney Møllers Fond
til Almene Formaal (NVK), Swedish Medical
Association (MO), Aarhus University (NVK), and
Aarhus University Hospital (NVK) funded this
study.

**Competing interests:** The authors have declared
that no competing interests exist.

## Conclusion

Higher levels of plasma free thiols at Day 1 and Day 5, which are reflective of lower levels of oxidative stress, are associated with better early and late graft function in recipients of a kidney graft from deceased donors.

## Trial registration

**ClinicalTrials.gov Identifier:** NCT01395719.

## Introduction

Renal transplantation can increase quality and length of life for patients with end-stage renal disease. Delayed graft function (DGF) mediated by ischaemia-reperfusion injury (IRI) is a major challenge particularly in deceased donor kidney transplantation [1], leading to the need for dialysis, higher risk of post-transplant complications, prolonged hospitalisation and inferior long term function in recipients of grafts from brain death donors [2, 3].

Oxidative stress, one of the most important contributors of the IRI process, may be involved in renal damage and DGF [4]. Oxidative stress can be considered as an imbalance between the production of reactive oxygen species (ROS) and the antioxidant capacity. Plasma thiols are known to scavenge reactive oxygen species, thereby protecting cells against oxidative stress. Although the greater part of plasma thiols is present in oxidized form [5, 6], the level of free thiols bears great relevance in determining the individual redox status. Systemic oxidative stress is associated with reduced levels of these circulating free thiols since these can be rapidly oxidised in conditions of increased production of ROS [7]. Increased ROS production, as reflected by decreased levels of free thiols, is often observed in human diseases in which oxidative stress plays a pathophysiological role, such as diabetes, chronic kidney disease, heart failure, cancer and Crohn's disease [8–10]. This indicates that the circulating level of free thiols directly reflects the systemic redox status and free thiol groups are assumed to play a protective role against oxidative stress due to their potent ability to scavenge ROS [11]. In addition to low molecular weight molecules, most of redox-active thiol groups (approximately 75% of the total thiol pool) are present in blood proteins, mainly albumin [7]. We have previously demonstrated that higher systemic free thiol levels predict better graft survival and lower mortality in renal transplant patients [12].

In this study, we hypothesised that higher systemic levels of free thiols, representing less oxidative stress, are associated with a better early and late graft function in patients subjected to renal transplantation. Our aim was therefore to examine the changes in systemic free thiols following deceased donor kidney transplantation and to correlate this biomarker with DGF as well as early and late graft function.

## Materials and methods

### Study design

This present study is a post-hoc analysis including patients from the multinational, double-blinded, randomised, controlled trial CONTEXT (ClinicalTrials.gov Identifier: NCT01395719) [13]. The CONTEXT study investigated the effect of remote ischaemic conditioning (RIC) in the recipient during kidney transplantation with deceased donors. The conditioning was performed by repetitive inflation of a cuff to occlude the blood supply to the leg on

the side not used for transplantation before reperfusion of the kidney graft. The study included 225 kidney transplant recipients from June 2011 to December 2014. Three patients were withdrawn from the study leaving 222 patients in the entire cohort. Two hundred patients received a graft from brain death donors whereas 22 received grafts from donors with circulatory death. The primary endpoint of the study was the estimated time to a 50% reduction in P-creatinine (tCr50) as a marker of early graft function [14]. The study was approved by relevant data protection agencies and ethical committees (Denmark: The National Committee on Health Research Ethics; Sweden: Regional Ethical Board; the Netherlands: METCUMCG) and performed in accordance with the Declaration of Helsinki.

## Inclusion

Patients receiving kidneys from deceased donors were included after informed consent in Aarhus, Denmark (n = 132); Gothenburg, Sweden (n = 46); and Rotterdam (n = 22) and Groningen (n = 25), the Netherlands. Informed consent was obtained from all patients after submission to hospital prior to transplantation [15]. The donation was organised by Scandia-Transplant or EuroTransplant with no connection between the donation process and transplant recipients.

Plasma free thiols were measured in a total of 217 patients who participated in the CON-TEXT trial [15]. Due to graft removal or primary non-function, eleven patients were excluded from the tCr50 analysis. Demographic and clinical information of the transplant recipients (age, gender, plasma (P-) creatinine, need for dialysis) was collected from hospital records.

## Ethics statement

None of the transplant donors was from a vulnerable population. No informed consent was obtained from the donors to participate in the CONTEXT study, but the organ donation was organized through ScandiaTransplant or EuroTransplant.

## Biochemical analyses

Blood samples were collected at baseline (n = 217), 30 minutes (n = 216) and 90 minutes (n = 206) after reperfusion of the kidney, Day 1 (n = 194) and Day 5 (n = 195) after transplantation. The samples were centrifuged and stored at -80˚C.

Plasma free thiols were measured as previously described [12]. Seventy-five μl plasma was diluted 1:4 in 0.1 M Tris buffer (pH 8.2) and transferred to 96-well plates. Using a Sunrise microplate reader (Tecan Trading AG, Männedorf, Switzerland), background absorption was measured at 412 nm with a reference filter at 630 nm. Subsequently, 10μl 3.8 mM 5,5′-Dithiobis (2-nitrobenzoic acid) (DTNB; Sigma Aldrich, Zwijndrecht, Netherlands) in 0.1 M phosphate buffer (pH 7) was added to the samples. After 20 min of incubation at room temperature, absorption was read again. The concentration of plasma free thiols in the samples was determined by comparing their absorbance readings to a standard calibration curve of L-cysteine (15–1000 μM; Fluka Biochemika, Buchs, Switzerland) in 0.1 M Tris and 10 mM EDTA (pH 8.2). Using this detection method, which has been thoroughly validated and has proven consistency across different centers, total free thiol content is measured, consisting of the combination of protein-bound free thiols and low-molecular-weight (LMW) free thiols (e.g., cysteine, homocysteine, and glutathione) [7, 12, 16].

P-creatinine was measured as part of the daily routine: twice daily the first week and twice weekly until 30 days after transplantation. If dialysis was needed post-transplant P-creatinine was measured twice a week until 30 days after last dialysis session.

## Outcome parameters

tCr50 was calculated as previously described [14]. DGF was defined as the need for dialysis within the first post-transplant week. Measured glomerular filtration rate (mGFR) was performed as $^{51}$chrome-ethylenediamine tetraacetic acid ($^{51}$Cr-EDTA) plasma clearance [17] on Day 5 (n = 89) and at twelve months (n = 137) after transplantation in patients included in Aarhus and Gothenburg with clinically detectable kidney graft function. The results were standardised to body surface.

## Statistical analyses

Demographic and clinical characteristics of the transplant recipients were presented as proportions $n$ (%), medians (interquartile range, IQR) or means (standard deviation, SD). Assessment of normality of distributions was performed using histograms. Simple linear regression was used to correlate the level of free thiols to kidney graft function while adjusting for recipient age and gender as relevant covariates and $r^2$ was adjusted for the degrees of freedom. mGFR at Day 5 and tCr50 were logarithmically transformed to obtain normal distributions–hence the estimates are presented as a doubling in mGFR Day 5 and tCr50, respectively. Multivariate repeated measurements ANOVA was used to compare the level of systemic free thiols between two groups (e.g. treatment and DGF). Student's T test was used to compare the individual timepoints in case the ANOVA revealed a difference. Stata® version 16 for Windows (StataCorp LP) was used to perform the statistical analyses.

# Results

## Recipient and donor characteristics

Table 1 shows the baseline characteristics of the 217 renal transplant recipients. Immunosuppression at discharge, original kidney disease, comorbidity and donor age and gender are all included in the table.

Free thiols at all time points followed a normal distribution and no transformation was needed for further analysis. Systemic free thiol levels were in accordance with previous studies that employed the same detection method [8, 9, 12, 16, 18]. In addition, no correlation was found between the time of baseline sampling and the level of free thiols (p = 0.76) (Fig 1), indicating stability of free thiols in the frozen samples. No difference was observed in the level of free thiols between recipients of kidneys from brain death donors and circulatory death donors except at Day 1 where the level of free thiols was higher among patients receiving a graft from a donor with circulatory death (p = 0.04).

We found no difference in plasma free thiol levels depending on treatment (RIC vs sham-RIC) at baseline, 30 minutes, 90 minutes, Day 1 or Day 5 after kidney transplantation (p = 0.25) (S1 Fig), and hence the data from the two groups were pooled for additional analysis.

## Free thiols and early graft function

Table 2 shows the associations between free thiols and initial graft function depicted as mGFR Day 5 and tCr50. Higher levels of free thiols measured only 90 mins after reperfusion are weakly correlated to a higher mGFR at Day 5. The association strengthens when free thiols are measured at Day 1 and Day 5 (Table 2 and Fig 2). A higher level of free thiols is weakly correlated to a shorter tCr50. Interestingly, the level of free thiols at baseline, 30 minutes and 90 minutes, but not Day 1, was significantly higher among patients experiencing DGF (Fig 3).

**Table 1. Recipient and donor characteristics.**

| | n = 219 |
|---|---|
| Recipient age (years) | 59.0 (49.4–66.0) |
| Recipient gender, female | 83 (38%) |
| Baseline P-free thiols (μM) | 363 (282–449) |
| Estimated baseline P-creatinine (μmol/l)[a] | 629 (498–762) |
| Immunosuppression at discharge | |
| Tacrolimus[b] | 200 (94%) |
| Mycophenolate mofetil | 212 (98%) |
| Corticosteroids | 206 (95%) |
| Original renal disease | |
| Glomerulopathy | 50 (23%) |
| ADPKD | 43 (20%) |
| Diabetes mellitus | 25 (12%) |
| Vascular/hypertension | 23 (11%) |
| Reflux/obstructive | 7 (3%) |
| Other | 23 (11%) |
| Unknown | 46 (21%) |
| Pre-transplant diabetes | 42 (19%) |
| Hypertension | 195 (90%) |
| Donor age (years) | 58 (52–65) |
| Donor gender, female | 99 (46%) |
| Cold ischemia time (hours)[a] | 13.5 (4.6) |

Values are presented as proportions *n* with corresponding percentages (%), medians (interquartile range) or means (standard deviation).

[a] n = 209 due to primary non-function or early graftectomy.

[b] n = 212 due to missing samples.

### Free thiols correlate with graft function at twelve months post-transplant

Higher levels of free thiols measured in the first hours and early days after kidney transplantation correlated with higher mGFR at twelve months (Table 3). Only weak correlations were found for free thiols measured at 30 and 90 minutes after reperfusion, whereas free thiols at Day 1 and Day 5 correlated moderately strong.

## Discussion

In the present study a greater increase in plasma free thiols, indicating a favourable redox balance during the first days after renal transplantation, correlated significantly with early kidney graft function (higher mGFR and shorter tCr50) suggesting that less oxidative stress associates with a better early graft function. Conversely, higher levels of free thiols at 30 minutes and 90 minutes after graft reperfusion were observed among patients experiencing DGF. Interestingly, we also identified that higher free thiols early after transplantation were associated with better one-year graft function. Taken together, these data suggest that the level of oxidative stress in the early phase after transplantation relates to graft onset and predicts kidney graft function at twelve months post-transplantation.

We did not observe any difference in free thiols as a result of the RIC procedure. The CONTEXT study was designed to test whether remote RIC could improve the outcome after renal transplantation. In line with the absence of detectable effects of RIC on free thiols, no effect of

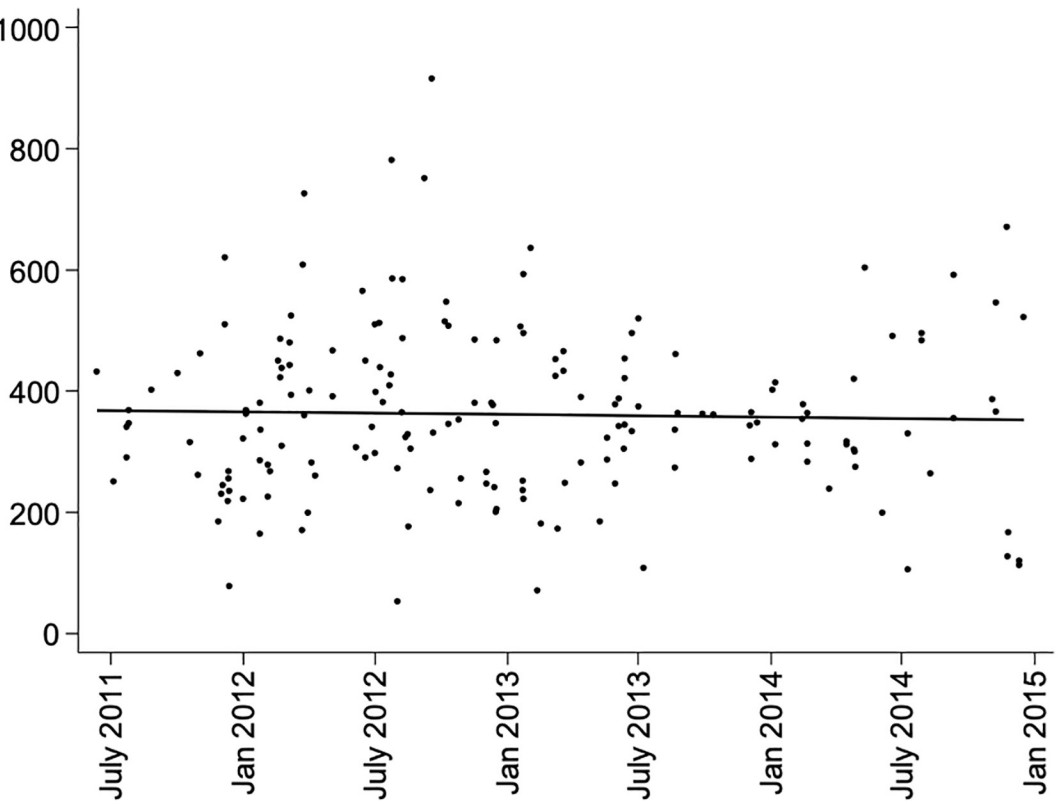

**Fig 1. Graph showing the level of free thiols (μM) measured at baseline over the total study period (p = 0.76).**

RIC on kidney function and other biomarkers as NGAL was observed in the CONTEXT study [15, 19, 20], therefore the analyses included in this study was performed on pooled data.

We have previously demonstrated that increased systemic levels of free thiols are associated with better graft survival and lower mortality in renal transplant recipients [12]. In addition, other studies have demonstrated that malondialdehyde (MDA), an end product of oxidative lipid peroxidation, predicts allograft survival and the possibility of DGF [21–23], again highlighting the relevance of oxidative stress in the early phase of renal transplantation.

Higher levels of plasma free thiols are a reflection of a beneficial systemic redox status since free thiols play a protective role against oxidative stress and act thereby as an independent

**Table 2. The correlations between free thiols at different time points and mGFR on Day 5 or tCr50.**

| Time point of free thiols sampling | | | | mGFR Day 5 | | | | tCr50 | | | | | |
|---|---|---|---|---|---|---|---|---|---|---|---|---|---|
| | n | $\beta^a$ | 95% CI$^a$ | p | $r^2_{adj.}$ | $p^b$ | $r^2_{adj.}{}^b$ | n | $\beta^a$ | 95% CI$^a$ | p | $r^2_{adj.}$ | $p^b$ | $r^2_{adj.}{}^b$ |
| 30 minutes | 89 | 33 | (2;65) | 0.04 | 0.04 | 0.09 | 0.02 | 206 | 6 | (-3;16) | 0.20 | 0.003 | 0.15 | 0.004 |
| 90 minutes | 86 | 42 | (10;75) | 0.01 | 0.06 | 0.03 | 0.04 | 196 | 1 | (-9;11) | 0.84 | -0.005 | 0.70 | -0.006 |
| Day 1 | 78 | 63 | (32;94) | <0.001 | 0.16 | <0.001 | 0.14 | 188 | -11 | (-22;-1) | 0.03 | 0.02 | 0.03 | 0.02 |
| Day 5 | 87 | 73 | (46;99) | <0.001 | 0.25 | <0.001 | 0.24 | 191 | -20 | (-29;-11) | <0.001 | 0.08 | <0.001 | 0.08 |

[a]standardised beta coefficients corresponding to a doubling in mGFR or tCr50, respectively.

[b]adjusted for recipient age and sex. CI = confidence interval. $r^2_{adj.}$ = correlation coefficient.

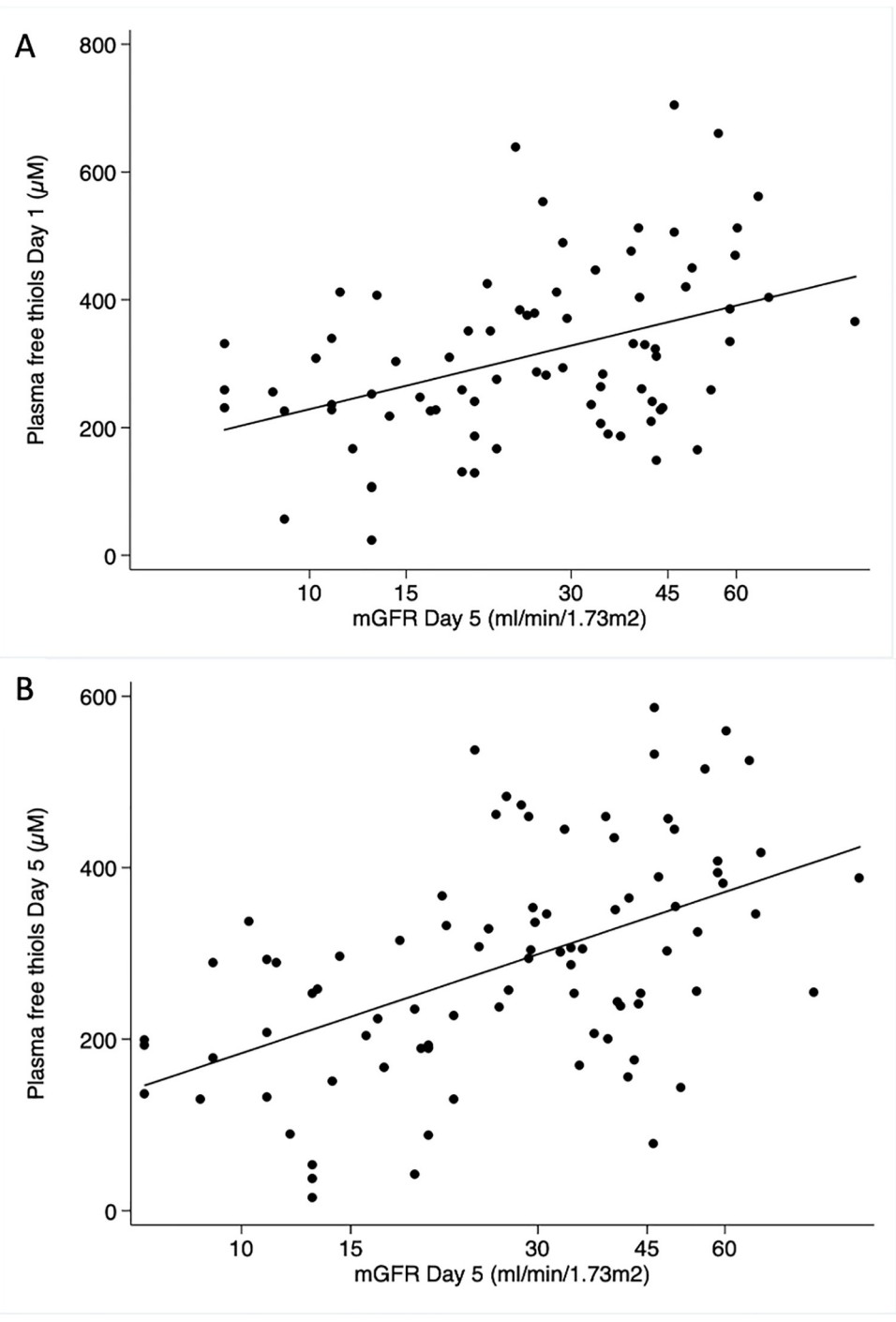

**Fig 2. Simple linear regression of plasma free thiols and mGFR at Day 5.** A) Free thiols at Day 1. B) Free thiols at Day 5.

marker of ROS levels [24]. Given that free thiol levels correlate with GFR we cannot exclude the possibility, that the association between the early increase in plasma free thiol levels and shorter tCr50 is, at least in part, a reflection of a greater GFR rather than an independent marker of ROS. However, since oxidative stress is closely related to IRI, which is an established

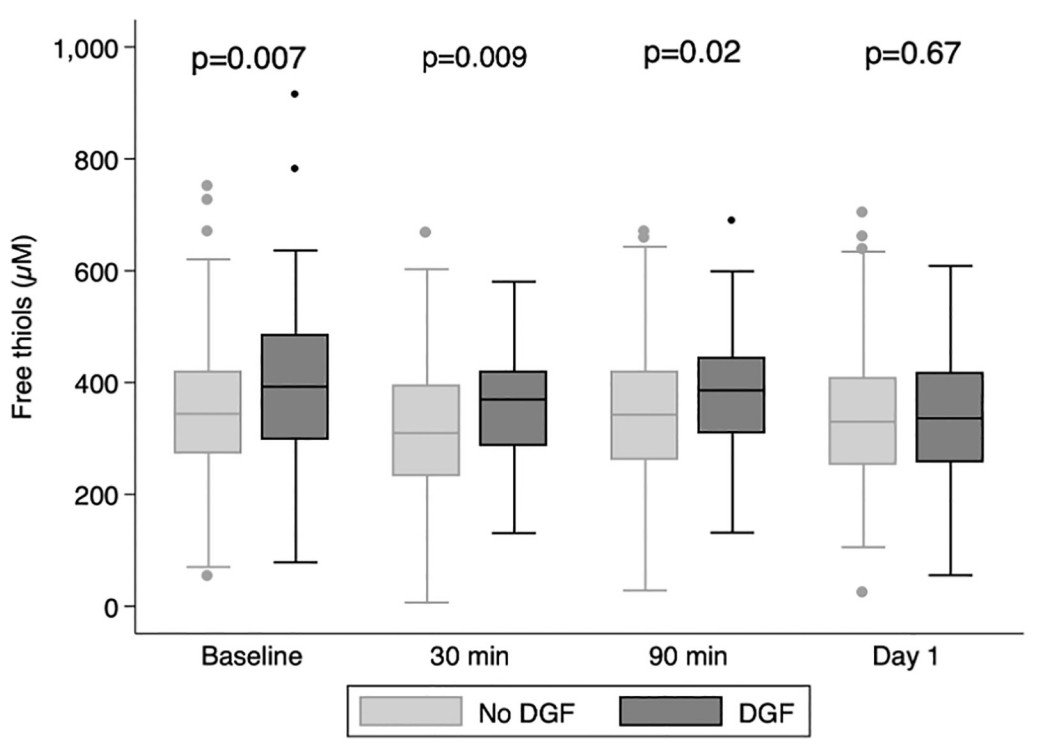

**Fig 3. Box plot of the level of free thiols (μM) at different time points and delayed graft function (DGF).**

complication of renal transplantation, an association between free thiols and kidney damage due to ROS in the context of renal transplantation is conceivable. Furthermore, intrarenal ROS may affect both the glomerular filtration barrier [25] and intrarenal haemodynamics [26] providing additional rationale for the relation between free thiols and mGFR.

We observed higher levels of free thiols at 30 minutes and 90 minutes after graft reperfusion among patients experiencing DGF, which would suggest that the potential detrimental effects of low free thiols and less favourable redox status is not associated with a higher incidence of DGF. The correlation is weak and cannot be observed at Day 1, and thus the significance is uncertain. It is possible that this early increase in free thiols after graft reperfusion is related to an interaction with substances released intraoperatively, such as residual sulfhydryl groups groups from the kidney storage solution, which are not cleared as efficiently in case of DGF.

**Table 3. The correlations between free thiols at early time points and mGFR at twelve months.**

| Time point of free thiols sampling | mGFR at twelve months | | | | | | |
|---|---|---|---|---|---|---|---|
| | n | β[a] | 95% CI | p | $r^2_{adj.}$ | P[b] | $r^2_{adj.}$[b] |
| 30 minutes | 137 | 1.8 | (0.8;2.9) | 0.001 | 0.07 | 0.03 | 0.14 |
| 90 minutes | 130 | 1.9 | (0.8;2.9) | 0.001 | 0.08 | 0.02 | 0.14 |
| Day 1 | 124 | 2.9 | (1.9;3.9) | <0.001 | 0.20 | <0.001 | 0.23 |
| Day 5 | 132 | 2.5 | (1.5;3.5) | <0.001 | 0.16 | <0.001 | 0.20 |

[a]standardised beta coefficient.

[b]adjusted for recipient age and sex. CI = confidence interval. $r^2_{adj.}$ = correlation coefficient.

The study is strengthened by the relatively large number of unselected deceased donor kidney transplant recipients. Moreover, the study is a large multicenter study carried out in Denmark, Sweden and the Netherlands. On the other hand, we recognise certain limitations of the study. Given the fact that the majority of patients were Caucasians, the generalisability of our results in subjects with other ethnicities remains unknown and our study did not prove any causal relationship. Furthermore, the absence of data on plasma albumin or total plasma protein levels refrained us from the possibility to adjust total free thiol levels by calculating the free thiol or total protein/albumin ratio. This protein adjustment would have enabled us to indirectly account for total thiol content as proteins harbour the largest amount of thiols and therefore quantitatively determine the amount of potentially detectable free thiols [16]. However, previous studies from our lab have demonstrated that in most cases, this protein adjustment does not severely affect the eventual results obtained, and thus it does not always lead to different conclusions. Finally, no sufficient biomaterials were available in the present study to perform additional experiments in order to further characterise the thiol redox metabolome in our patients. Measurements of individual thiol species (e.g. cysteine, homocysteine and glutathione) or inclusion of extra indices (e.g., the protein thiolation index, PTI) could have provided us with more in-depth information on extracellular free thiol dynamics [27]. Future studies could be designed to focus on a combination of key components of the thiol redox metabolome resulting in an integrative biomarker approach, representing multiple redox-regulated metabolic pathways. However, such "redox metabolomics" approaches are still under development and are accompanied by various methodological constraints. Similarly, it is as yet unclear what criteria potential thiol redox biomarkers should fulfil to be reliably reflective of the human redox system in a high-throughput setting [28]. Therefore, the single quantification of total free thiol content in serum or plasma is currently considered one of the most useful screening tools for measuring the whole-body redox status in clinical and translational studies.

## Conclusion

In conclusion, this study suggests that plasma free thiols correlate with early graft function, and can predict kidney function at twelve months post-transplantation.

## Supporting information

**S1 Fig. The level of plasma free thiols at different time points depending on treatment (Sham-RIC vs. RIC).** RIC = remote ischaemic conditioning.
(DOCX)

## Author Contributions

**Conceptualization:** Bente Jespersen, Henrik Birn, Nicoline V. Krogstrup, Henri G. D. Leuvenink, Harry van Goor, Rikke Nørregaard.

**Data curation:** Marie B. Nielsen, Nicoline V. Krogstrup, Harry van Goor.

**Formal analysis:** Marie B. Nielsen.

**Funding acquisition:** Marie B. Nielsen, Nicoline V. Krogstrup.

**Methodology:** Marie B. Nielsen, Bente Jespersen, Henrik Birn, Nicoline V. Krogstrup, Harry van Goor, Rikke Nørregaard.

**Supervision:** Harry van Goor, Rikke Nørregaard.

**Writing – original draft:** Marie B. Nielsen, Harry van Goor, Rikke Nørregaard.

**Writing – review & editing:** Marie B. Nielsen, Bente Jespersen, Henrik Birn, Nicoline V. Krogstrup, Arno R. Bourgonje, Henri G. D. Leuvenink, Harry van Goor, Rikke Nørregaard.

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
