## [Decision Letter · Decision Letter 0]

10 Dec 2020

PONE-D-20-32900

Elevated plasma free thiols are associated with early and one-year graft function in renal transplant recipients

PLOS ONE

Dear Dr. Nielsen,

Thank you for submitting your manuscript to PLOS ONE. After careful consideration, we feel that it has merit but does not fully meet PLOS ONE’s publication criteria as it currently stands. Therefore, we invite you to submit a revised version of the manuscript that addresses the points raised during the review process.

We look forward to receiving your revised manuscript.

Kind regards,

Gianpaolo Reboldi, MD, MSc, PhD

Academic Editor

PLOS ONE

Journal Requirements:

2. Please note that PLOS does not permit references to “data not shown.” Authors should provide the relevant data within the manuscript, the Supporting Information files, or in a public repository. If the data are not a core part of the research study being presented, we ask that authors remove any references to these data.

3. In your Methods section, please provide additional information about the participant recruitment method and the demographic details of your participants. Please ensure you have provided sufficient details to replicate the analyses such as: a) a description of how participants were recruited, and b) descriptions of where participants were recruited and where the research took place.

4. We note that your study involved tissue/organ transplantation. Please provide the following information regarding tissue/organ donors for transplantation cases analyzed in your study.

1. Please provide the source(s) of the transplanted tissue/organs used in the study, including the institution name and a non-identifying description of the donor(s).

2. Please state in your response letter and ethics statement whether the transplant cases for this study involved any vulnerable populations; for example, tissue/organs from prisoners, subjects with reduced mental capacity due to illness or age, or minors.

- If a vulnerable population was used, please describe the population, justify the decision to use tissue/organ donations from this group, and clearly describe what measures were taken in the informed consent procedure to assure protection of the vulnerable group and avoid coercion.

- If a vulnerable population was not used, please state in your ethics statement, “None of the transplant donors was from a vulnerable population and all donors or next of kin provided written informed consent that was freely given.”

3. In the Methods, please provide detailed information about the procedure by which informed consent was obtained from organ/tissue donors or their next of kin. In addition, please provide a blank example of the form used to obtain consent from donors, and an English translation if the original is in a different language.

4. Please indicate whether the donors were previously registered as organ donors. If tissues/organs were obtained from deceased donors or cadavers, please provide details as to the donors’ cause(s) of death.

5. Please discuss whether medical costs were covered or other cash payments were provided to the family of the donor. If so, please specify the value of this support (in local currency and equivalent to U.S. dollars).

5. We noted in your submission details that a portion of your manuscript may have been presented or published elsewhere.

"Table 1 presenting recipient and donor characteristics (except free thiols level) have been published in previous CONTEXT study papers."

Please clarify whether this publication was peer-reviewed and formally published. If this work was previously peer-reviewed and published, in the cover letter please provide the reason that this work does not constitute dual publication and should be included in the current manuscript.

6. We note that you have indicated that data from this study are available upon request. PLOS only allows data to be available upon request if there are legal or ethical restrictions on sharing data publicly. For information on unacceptable data access restrictions, please see http://journals.plos.org/plosone/s/data-availability#loc-unacceptable-data-access-restrictions.

Reviewers' comments:

Reviewer's Responses to Questions

**Comments to the Author**

1. Is the manuscript technically sound, and do the data support the conclusions?

Reviewer #1: Partly

Reviewer #2: No

2. Has the statistical analysis been performed appropriately and rigorously? 

Reviewer #1: No

Reviewer #2: No

3. Have the authors made all data underlying the findings in their manuscript fully available?

Reviewer #1: No

Reviewer #2: Yes

4. Is the manuscript presented in an intelligible fashion and written in standard English?

Reviewer #1: Yes

Reviewer #2: Yes

5. Review Comments to the Author

Reviewer #1: The paper consists of a post hoc exploratory analysis of data generated from the randomized controlled "CONTEXT" trial to understand whether higher levels of free thiols is associated with better graft functions in renal transplants. I have some questions and clarifications, mainly from the statistical analysis side.

1. The statistical analyses plan lacks any power/sample size considerations. The study would improve, if some statements were provided in this regard, like what was the power the authors initially expected, for the samples available. They may use a desired statistical test at 5% significance level.

2. Statistical Analyses subsection written poorly, and analysis conducted is not up to the desired mark.

(a) "Simple linear regression was used to correlate continuous variables" doesn't have a clear meaning. Say something like, "A simple linear regression was used to assess the effect of XX on YYY, controlling for ZZZ (the confounders, etc).

(b) I do not understand the title of the Tables 2 and 3. Linear regression of response (Y) on the desired covariates (X) present the parameter estimates, standard errors, 95% confidence intervals, and p-values. Those need to be presented, if , at all, a linear regression was conducted. What is presented looks like some standard (adjusted) correlations.

(c) Looks like the data collection plan of thiol levels was longitudinal, like baseline, 30 and 90 minutes, etc. So, in addition to analyses at separate points and presenting correlations, I wonder why a formal longitudinal data analyses was not conducted, via linear mixed models, or generalized estimating equations? The time points of thiols sampling could enter the model to model the time trend, or something like that.

Reviewer #2: The manuscript “Elevated plasma free thiols are associated with early and one-year graft function in renal transplant recipients” by Nielsen et al. fosters the hypothesis that “higher (i.e. less oxidated) levels of free thiols as a biomarker of reduced oxidative stress are associated with a better initial graft function or a higher GFR.” (last sentence of the first paragraph of the abstract).

I have several concerns on this manuscript, the main ones are:

1. the levels of free thiols in plasma reported in this study appear to be much higher than expected (between 100 and 700 uM). The analysis protocol described in the manuscript should lead to measure a minor fraction of these compounds present in this and in other extracellular fluids, i.e. the free (or reduced) form of total thiols only. In fact, such protocol does not include reducing agents to break the disulfide bridges of the oxidized forms, which is a prerequisite for the reaction of thiols with DTND. Therefore the levels expected from this assay should be much lower. Please note that the large majority of total thiols in plasma and other extracellular fluids is present under the oxidized form, which cannot be measured with the proposed assay (i.e. as disulfides or mixed disulfides with Cys34 of Albumin, with an average ratio of reduced to oxidized forms of 0.2) (Giustarini, Dalle-Donne et al. 2006; Jones and Liang 2009; Galli et al. Free Rad Res, 2014; Galli F. et al Kideny Int 2013). Therefore, I do not know what exactly the authors have measured in this study, and this concerns me a lot because the uncertainty on the proposed results invalidates the study hypothesis and aim (i.e. to expect changes in the levels of the reduced form of plasma thiols that may reflect their oxidation state and then the presence of oxidative stress in the transplanted patients).

2. To demonstrate the presence of an altered redox of the extracellular environment and to link this alteration with the function of specific organs, more experiments should be performed and other laboratory indices must be investigated.

a. First, the different thiol/disulfide couples of the different low-molecular mass thiols (Cys, Hcy, Cys-Gly, GSH) of extracellular fluids should be investigated together with protein S-thiolation and blood cell thiols. Please note that in healthy subjects these are in the following ranges: 150-300 uM total Cys (including that coming from CySS and Cys-Gly), while Hcy is approx. 5-10 uM and GSH is usually 2-6 uM) and the mean ratio of reduced to oxidized forms is 0.2. In CKD patients on standard hemodialysis the absolute levels of total thiols significantly increase with different extent of modification in the individual thiol species (Galli et al. Free Rad Res, 2014). The identification of these subclasses is much more informative compared with that proposed in the present manuscript.

b. Second, serum albumin and the levels of its thiolation (mixed disulfides) should be considered to explain the interindividual differences observed in this study (see figure 1 and 2). For example, Cys is largely engaged in mixed disulfide formation and Hcy is more than 75 % bound to serum albumin (Galli F. et al Kideny Int 2013; Galli et al. Free Rad Res, 2014).

c. Protein thiolation in plasma is a relevant indicator of oxidative stress in age-related and inflammatory diseases, including CKD (Reggiani et al. 2015, Fanti, Giustarini et al. 2015), and increased levels of biomarker linearly correspond to the decline of thiol to disulfide balance in extracellular fluids. This is relevant biomarker to utilize if one would like to explore the impaired redox of a patient with a systemic (not organ-specific) approach. This biomarker should be investigated together with other indices of damage of plasma proteins and/or polyunsaturated lipids (e.g. protein carbonylation, lipid peroxidation products, etc.). May be the Authors have a bank of samples with aliquots of plasma still available for these determinations.

d. Organ-specific indications cannot be expected from the proposed laboratory strategy to explore plasma thiols. The investigation of individual thiol species would provide much higher chances to obtain some level of information on the transplanted organ (see later in the next point).

3. The changes observed in the levels of free thiols in plasma of this study and their correlation with the success of transplantation and organ function are more than expected if we consider that tubular epithelial cells are very rich in gamma-glutamyl transpeptidase or ��GT (Giustarini, Galvagni et al. 2020). Therefore, reduced or absent function of tubular epithelia cells observed in the late stages of kidney disease, is expected to impair the renal metabolism and extracellular levels of LMW thiols, and especially of Cys. Possibly, what the Authors in this study are measuring with their thiol assay in plasma is the ��GT activity of the transplanted organ that obviously is higher in successfully treated subjects.

4. The redox balance of extracellular thiols declines with the subject’ age (Jones, Mody et al. 2002, Giustarini, Dalle-Donne et al. 2006) and such a decline is even more rapid in case of of premature aging and impaired redox homeostasis, which are characteristic conditions of CKD (Galli et al. Free Rad Res, 2014; Reggiani et al. 2015, Fanti, Giustarini et al. 2015). The results in this study (when the actual reduced form of thiols will be measured) should be corrected for the age of the patients as potential confounding factor.

5. Based on the correlation between thiols and mGFR it could be assumed that GFR could be utilized instead of thiols as a biomarker of a successful transplantation, which I guess is routine in the clinical monitoring of transplanted patients. What plasma thiols (those measured with this study) actually add up to the already available indices of organ function in transplantation protocols?

6. The term “oxidated” should be revised and substituted with oxidized.

7. The Authors have disregarded most of the studies performed so far on plasma thiols in the introduction of their study and in the discussion of the results. This and other aspects discussed earlier in this revision report, demonstrate poor confidence with this topic. I suggest to refer to experts in the field of redox biology and medicine, and especially in thiol analysis, to obtain sufficed advise during the revision of their manuscript.

6. PLOS authors have the option to publish the peer review history of their article (what does this mean?). If published, this will include your full peer review and any attached files.

Reviewer #1: No

Reviewer #2: **Yes: **Francesco Galli

---

## [Author Response · Author response to Decision Letter 0]

11 Feb 2021

Rebuttal letter

Our responses are included in red. Lines numbers refer to Manuscript with Tracked Changes

https://journals.plos.org/plosone/s/file?id=wjVg/PLOSOne_formatting_sample_main_body.pdfand

The headings and subheadings have been reedited.

The author affiliations have been reedited.

2. Please note that PLOS does not permit references to “data not shown.” Authors should provide the relevant data within the manuscript, the Supporting Information files, or in a public repository. If the data are not a core part of the research study being presented, we ask that authors remove any references to these data.

The data is now attached to the manuscript as a supplemental figure.

3. In your Methods section, please provide additional information about the participant recruitment method and the demographic details of your participants. Please ensure you have provided sufficient details to replicate the analyses such as: a) a description of how participants were recruited, and b) descriptions of where participants were recruited and where the research took place.

Participant recruitment as well as number of patients from each center has been added to the revised manuscript in the Materials and Methods section (lines 84-87). For more information see Krogstrup et al, AJT, 2016 (reference no. 15). Age and gender of recipients are presented in Table 1.

Krogstrup N V., Oltean M, Nieuwenhuijs-Moeke GJ, Dor FJMF, Møldrup U, Krag SP, et al. Remote Ischemic Conditioning on Recipients of Deceased Renal Transplants Does Not Improve Early Graft Function: A Multicenter Randomized, Controlled Clinical Trial. Am J Transplant. 2016;1–8. doi: 10.1111/ajt.14075

4. We note that your study involved tissue/organ transplantation. Please provide the following information regarding tissue/organ donors for transplantation cases analyzed in your study.

1. Please provide the source(s) of the transplanted tissue/organs used in the study, including the institution name and a non-identifying description of the donor(s).

The organs were provided from deceased donors (after brain death or circulatory death). The organ donation was organised via ScandiaTransplant or EuroTransplant, this has now been added to the revised manuscript in the Materials and Methods section (lines 87-88).

Donor age and gender are presented in Table 1. 

2. Please state in your response letter and ethics statement whether the transplant cases for this study involved any vulnerable populations; for example, tissue/organs from prisoners, subjects with reduced mental capacity due to illness or age, or minors.

Please see the cover letter (ethics statement).

- If a vulnerable population was used, please describe the population, justify the decision to use tissue/organ donations from this group, and clearly describe what measures were taken in the informed consent procedure to assure protection of the vulnerable group and avoid coercion.

- If a vulnerable population was not used, please state in your ethics statement, “None of the transplant donors was from a vulnerable population and all donors or next of kin provided written informed consent that was freely given.”

3. In the Methods, please provide detailed information about the procedure by which informed consent was obtained from organ/tissue donors or their next of kin. In addition, please provide a blank example of the form used to obtain consent from donors, and an English translation if the original is in a different language.

In the Materials and Methods section the following has been added: “Informed consent was obtained from all patients after submission to hospital prior to transplantation (15). The donation was organised by ScandiaTransplant or EuroTransplant with no connection between the donation process and transplant recipients.” (lines 86-88)

The deceased donor donation occurred completely independent of this project that only involved the transplant recipients. Therefore, donor families were only asked whether they would consent that their loved ones could be donors according to the usual routines in the three countries, where donation was organised by ScandiaTransplant or EuroTransplant.

4. Please indicate whether the donors were previously registered as organ donors. If tissues/organs were obtained from deceased donors or cadavers, please provide details as to the donors’ cause(s) of death.

It was not registered in the CONTEXT study whether the organ donors were previously registered as organ donors or whether the decision was made by next of kin. The donor’s cause(s) of death has previously been published (Krogstrup et al, AJT, 2016).

Krogstrup N V., Oltean M, Nieuwenhuijs-Moeke GJ, Dor FJMF, Møldrup U, Krag SP, et al. Remote Ischemic Conditioning on Recipients of Deceased Renal Transplants Does Not Improve Early Graft Function: A Multicenter Randomized, Controlled Clinical Trial. Am J Transplant. 2016;1–8. doi: 10.1111/ajt.14075

5. Please discuss whether medical costs were covered or other cash payments were provided to the family of the donor. If so, please specify the value of this support (in local currency and equivalent to U.S. dollars).

No medical cost or cash payments were provided to the family of the donor.

5. We noted in your submission details that a portion of your manuscript may have been presented or published elsewhere.

"Table 1 presenting recipient and donor characteristics (except free thiols level) have been published in previous CONTEXT study papers."

Please clarify whether this publication was peer-reviewed and formally published. If this work was previously peer-reviewed and published, in the cover letter please provide the reason that this work does not constitute dual publication and should be included in the current manuscript.

Please see the cover letter (previous publications).

6. We note that you have indicated that data from this study are available upon request. PLOS only allows data to be available upon request if there are legal or ethical restrictions on sharing data publicly. For information on unacceptable data access restrictions, please see http://journals.plos.org/plosone/s/data-availability#loc-unacceptable-data-access-restrictions.

Please see the cover letter (data availability).

Please see the cover letter (data availability).

5. Review Comments to the Author

Reviewer #1: The paper consists of a post hoc exploratory analysis of data generated from the randomized controlled "CONTEXT" trial to understand whether higher levels of free thiols is associated with better graft functions in renal transplants. I have some questions and clarifications, mainly from the statistical analysis side.

1. The statistical analyses plan lacks any power/sample size considerations. The study would improve, if some statements were provided in this regard, like what was the power the authors initially expected, for the samples available. They may use a desired statistical test at 5% significance level.

We thank the reviewer for this important comment. Before executing the CONTEXT study, a power calculation on the primary endpoint was performed in order to find the number of patients to include in the treatment arms (Krogstrup et al., BMJ Open, 2015 – reference no. 13). However, as this study is a post hoc analysis, no power calculation was performed.

Krogstrup N V, Oltean M, Bibby BM, Nieuwenhuijs-Moeke GJ, Dor FJMF, Birn H, et al. Remote ischaemic conditioning on recipients of deceased renal transplants, effect on immediate and extended kidney graft function: a multicentre, randomised controlled trial protocol (CONTEXT). BMJ Open. 2015;5(8):e007941.

2. Statistical Analyses subsection written poorly, and analysis conducted is not up to the desired mark.

(a) "Simple linear regression was used to correlate continuous variables" doesn't have a clear meaning. Say something like, "A simple linear regression was used to assess the effect of XX on YYY, controlling for ZZZ (the confounders, etc).

Thank you for the suggestion to improve the statistical subsection. It has now been rewritten in the revised manuscript in the Materials and Methods section: 

“Simple linear regression was used to correlate the level of free thiols to kidney graft function while adjusting for recipient age and gender as relevant covariates and r2 was adjusted for the degrees of freedom.” (lines 118-120)

(b) I do not understand the title of the Tables 2 and 3. Linear regression of response (Y) on the desired covariates (X) present the parameter estimates, standard errors, 95% confidence intervals, and p-values. Those need to be presented, if , at all, a linear regression was conducted. What is presented looks like some standard (adjusted) correlations.

We agree that the titles may cause some confusion. For that purpose we have now changed the titles of Table 2 and 3 as well as added the standardised beta coefficient with 95% confidence intervals of the linear regression analyses (Table 2 and 3). 

(c) Looks like the data collection plan of thiol levels was longitudinal, like baseline, 30 and 90 minutes, etc. So, in addition to analyses at separate points and presenting correlations, I wonder why a formal longitudinal data analyses was not conducted, via linear mixed models, or generalized estimating equations? The time points of thiols sampling could enter the model to model the time trend, or something like that.

The data collection plan was indeed longitudinal of origin. We appreciate the reviewer’s comment since it touches upon a fair argument that it is appropriate to use a formal longitudinal data analysis method. Therefore, the comparison between the two groups (RIC vs. sham-RIC as well as DGF vs. no DGF) has now been performed using multivariate repeated measurements ANOVA. The method is described in the statistics section in the Material and Methods (lines 122-124) and the result of the RIC vs sham-RIC is included in line 148. 

Reviewer #2: The manuscript “Elevated plasma free thiols are associated with early and one-year graft function in renal transplant recipients” by Nielsen et al. fosters the hypothesis that “higher (i.e. less oxidated) levels of free thiols as a biomarker of reduced oxidative stress are associated with a better initial graft function or a higher GFR.” (last sentence of the first paragraph of the abstract).

I have several concerns on this manuscript, the main ones are:

1. the levels of free thiols in plasma reported in this study appear to be much higher than expected (between 100 and 700 uM). The analysis protocol described in the manuscript should lead to measure a minor fraction of these compounds present in this and in other extracellular fluids, i.e. the free (or reduced) form of total thiols only. In fact, such protocol does not include reducing agents to break the disulfide bridges of the oxidized forms, which is a prerequisite for the reaction of thiols with DTND. Therefore the levels expected from this assay should be much lower. Please note that the large majority of total thiols in plasma and other extracellular fluids is present under the oxidized form, which cannot be measured with the proposed assay (i.e. as disulfides or mixed disulfides with Cys34 of Albumin, with an average ratio of reduced to oxidized forms of 0.2) (Giustarini, Dalle-Donne et al. 2006; Jones and Liang 2009; Galli et al. Free Rad Res, 2014; Galli F. et al Kideny Int 2013). Therefore, I do not know what exactly the authors have measured in this study, and this concerns me a lot because the uncertainty on the proposed results invalidates the study hypothesis and aim (i.e. to expect changes in the levels of the reduced form of plasma thiols that may reflect their oxidation state and then the presence of oxidative stress in the transplanted patients).

We thank the reviewer for outlining the above concerns. It is correct that the true redox potential of plasma thiols is difficult to address, but we are more than happy to further clarify the measurement method that was used in this study. Here, we have determined total free thiol content in plasma by derivatization with DTNB as thiol-reactive agent. This is based on the standardised Ellman reaction. Using this method, we actually measure the combination of protein-bound free thiols (since proteins were not removed from the samples) and low-molecular-weight (LMW) free thiols, e.g. cysteine, glutathione and homocysteine. The reviewer is correct that total thiol content (including oxidized thiols with disulphide bridges) was not measured here. In that case, we would have required to use a stronger reducing agent like dithiothreitol (DTT), which is able to additionally reduce disulphide bonds and oxidized protein-bound thiols. (Sutton TR et al., Redox Biol 2018).

In the present study, we determined total free thiol content since this method has been well-developed and optimized in the literature (Ellman reaction) and in our center. Our measurement method has been well validated. We did extensive tests to show that the samples are in optimal conditions. E.g. repeated freezing and thawing had no effect on the level of free thiols measured. The method has also proven its power to predict clinical parameters in many previous studies, in which free thiol levels were also in agreement with the range within the present study. (Frenay AS et al., Free Radic Biol Med 2016; Koning AM et al., Pharmacol Res 2016; Bourgonje AR et al., Front Physiol 2019; Abdulle AE et al., Physiol Rep 2019) In these studies, free thiols were also considered a reliable reflection of the systemic redox status, whereas total thiol content is usually referred to as being reflective of the systemic redox reserve. (Cortese-Krott MM et al., Antioxid Redox Signal 2017) Alternatively, one could define systemic redox status by measuring the ratio of reduced over oxidized thiols or by adjusting free thiol to total plasma protein content. In the latter way, you could indirectly account for total thiol content as proteins constitute by far the largest pool of redox-active thiols both in blood and in cells. Unfortunately, however, protein levels were not available for the present study and we do not have sufficient plasma left to add these measurements, but we can reassure the reviewer about this point as previous studies from our center also demonstrated no differences in results and conclusions with or without adjustment to total protein levels. 

Sutton TR, Minnion M, Barbarino F, et al. A robust and versatile mass spectrometry platform for comprehensive assessment of the thiol redox metabolome. Redox Biol 2018;16:359-380.

Frenay AS, de Borst MH, Bachtler M, Tschopp N, Keyzer CA, van den Berg E, et al. Serum free sulfhydryl status is associated with patient and graft survival in renal transplant recipients. Free Radic Biol Med 2016;99:345-351.

Koning AM, Meijers WC, Pasch A, Leuvenink HGD, Frenay AS, Dekker MM, et al. Serum free thiols in chronic heart failure. Pharmacol Res 2016;111:452-458.Koning AM et al., Pharmacol Res 2016

Bourgonje AR, Gabriëls RY, de Borst MH, Bulthuis MLC, Faber KN, van Goor H, et al. Serum Free Thiols Are Superior to Fecal Calprotectin in Reflecting Endoscopic Disease Activity in Inflammatory Bowel Disease. Antioxidants (Basel) (2019) 8(9):351.Bourgonje AR et al., Front Physiol 2019; 

Abdulle AE, Bourgonje AR, Kieneker LM, Koning AM, la Bastide-van Gemert S, Bulthuis MLC, et al. Serum free thiols predict cardiovascular events and all-cause mortality in the general population: a prospective cohort study. BMC Med (2020) 18(1):130.

Damba T, Bourgonje AR, Abdulle AE, Pasch A, Sydor S, van den Berg EH, et al. Oxidative stress is associated with suspected non-alcoholic fatty liver disease and all-cause mortality in the general population. Liver Int 2020;40(9):2148-2159.

Cortese-Krott MM, Koning A, Kuhnle GGC, Nagy P, Bianco CL, Pasch A, et al. The Reactive Species Interactome: Evolutionary Emergence, Biological Significance, and Opportunities for Redox Metabolomics and Personalized Medicine. Antioxid Redox Signal (2017) 27(10) :684-712.

2. To demonstrate the presence of an altered redox of the extracellular environment and to link this alteration with the function of specific organs, more experiments should be performed and other laboratory indices must be investigated.

a. First, the different thiol/disulfide couples of the different low-molecular mass thiols (Cys, Hcy, Cys-Gly, GSH) of extracellular fluids should be investigated together with protein S-thiolation and blood cell thiols. Please note that in healthy subjects these are in the following ranges: 150-300 uM total Cys (including that coming from CySS and Cys-Gly), while Hcy is approx. 5-10 uM and GSH is usually 2-6 uM) and the mean ratio of reduced to oxidized forms is 0.2. In CKD patients on standard hemodialysis the absolute levels of total thiols significantly increase with different extent of modification in the individual thiol species (Galli et al. Free Rad Res, 2014). The identification of these subclasses is much more informative compared with that proposed in the present manuscript.

We agree with the reviewer that from a redox biology perspective, it would be highly interesting to look into changes in different thiol/disulphide couples of different LMW thiols. Indeed, when investigating changes in free thiol levels, individual thiol species may be differentially affected by a certain clinical event and this may become clearer when performing the above suggested measurements. Although we would have strongly considered to perform these experiments when we had sufficient biomaterials left, this is currently outside the scope of the present manuscript. Our aim was to associate total free thiol content with early and one-year graft function in patients subjected to renal transplantation. Therefore, more in-depth analysis of thiol subclasses should be strongly considered for future studies to gather more information about the dynamics of thiol changes. (Discussion, lines 226-238)

b. Second, serum albumin and the levels of its thiolation (mixed disulfides) should be considered to explain the interindividual differences observed in this study (see figure 1 and 2). For example, Cys is largely engaged in mixed disulfide formation and Hcy is more than 75 % bound to serum albumin (Galli F. et al Kideny Int 2013; Galli et al. Free Rad Res, 2014).

Of course, we acknowledge the role of serum albumin in measuring total free thiol content. Ideally, we would have performed an adjustment to total protein or albumin level as albumin is quantitatively the most important human plasma protein, and plasma proteins harbour the largest amount of extracellular free thiols and therefore greatly determine the quantity of potentially detectable free thiol groups (Turell et al., Free Radic Biol Med, 2013). Unfortunately, in this study we had no albumin measurements available to be included in the analysis. Although we may be forced to accept this is a limitation of our study, we previously showed that adjustment had no effect on outcome (Abdulle et al., BMC Med, 2020; Bourgonje et al., Antioxidants, 2019). (Discussion, line 218-222)

Turell L, Radi R, Alvarez B. The thiol pool in human plasma: the central contribution of albumin to redox processes. Free Radic Biol Med. 2013 Dec;65:244–53.

Abdulle AE, Bourgonje AR, Kieneker LM, Koning AM, la Bastide-van Gemert S, Bulthuis MLC, et al. Serum free thiols predict cardiovascular events and all-cause mortality in the general population: a prospective cohort study. BMC Med (2020) 18(1):130.

Bourgonje AR, Gabriëls RY, de Borst MH, Bulthuis MLC, Faber KN, van Goor H, et al. Serum Free Thiols Are Superior to Fecal Calprotectin in Reflecting Endoscopic Disease Activity in Inflammatory Bowel Disease. Antioxidants (Basel) (2019) 8(9):351.

c. Protein thiolation in plasma is a relevant indicator of oxidative stress in age-related and inflammatory diseases, including CKD (Reggiani et al. 2015, Fanti, Giustarini et al. 2015), and increased levels of biomarker linearly correspond to the decline of thiol to disulfide balance in extracellular fluids. This is relevant biomarker to utilize if one would like to explore the impaired redox of a patient with a systemic (not organ-specific) approach. This biomarker should be investigated together with other indices of damage of plasma proteins and/or polyunsaturated lipids (e.g. protein carbonylation, lipid peroxidation products, etc.). 

First of all, we agree with the reviewer that protein thiolation is also a relevant indicator of oxidative stress in both healthy and diseased conditions and that additional measurements of other oxidative damage markers (e.g., protein carbonyls, lipid peroxidation products) would deliver us a more accurate reflection of the whole-body redox status. Of course, a combination of markers or combination of multiple read-outs may enhance reliability and applicability of the results. However, the measurement of total free thiols in plasma/serum actually behaves very similar as what the reviewer describes about protein thiolation: it is a relevant indicator of oxidative stress in age-related and inflammatory diseases and increased levels correspond to the decline of the thiol to disulphide balance in extracellular fluids as has been described in previous clinical studies that measured total free thiol content (Cortese-Krott MM et al., Antioxid Redox Signal 2017; Santolini et al., Curr Opin Physiol 2019; Cortese-Krott MM et al., 2020; Abdulle AE et al., BMC Med 2020; Damba T et al., Liver Int 2020; Frenay AS et al., Free Radic Bio Med 2016). All these studies supported the notion that one single quantification of total free thiols may be a robust method to evaluate the global redox state and accurately represent levels of systemic oxidative stress in a variety of conditions; for example, a previous study demonstrated an improved patient and graft survival in renal transplant recipients over several years of follow-up (Frenay AS et al, Free Radic Biol Med 2016).

Cortese-Krott MM, Koning A, Kuhnle GGC, Nagy P, Bianco CL, Pasch A, et al. The Reactive Species Interactome: Evolutionary Emergence, Biological Significance, and Opportunities for Redox Metabolomics and Personalized Medicine. Antioxid Redox Signal (2017) 27(10) :684-712.

Santolini J, Wootton SA, Jackson A, Feelisch M. The Redox Architecture of Physiological Function. Curr Opin Physiol 2019;9:34-47.

Cortese-Krott MM, Santolini J, Wootton SA, Jackson A, Feelisch M. The reactive species interactome. In: Sies H. Oxidative Stress: Eustress and Distress. Academic Press, 2020: 51-64.

Abdulle AE, Bourgonje AR, Kieneker LM, Koning AM, la Bastide-van Gemert S, Bulthuis MLC, et al. Serum free thiols predict cardiovascular events and all-cause mortality in the general population: a prospective cohort study. BMC Med (2020) 18(1):130.

Damba T, Bourgonje AR, Abdulle AE, Pasch A, Sydor S, van den Berg EH, et al. Oxidative stress is associated with suspected non-alcoholic fatty liver disease and all-cause mortality in the general population. Liver Int 2020;40(9):2148-2159.

Bourgonje AR, Gabriëls RY, de Borst MH, Bulthuis MLC, Faber KN, van Goor H, et al. Serum Free Thiols Are Superior to Fecal Calprotectin in Reflecting Endoscopic Disease Activity in Inflammatory Bowel Disease. Antioxidants (Basel) (2019) 8(9):351.

Frenay AS, de Borst MH, Bachtler M, Tschopp N, Keyzer CA, van den Berg E, et al. Serum free sulfhydryl status is associated with patient and graft survival in renal transplant recipients. Free Radic Biol Med 2016;99:345-351.

Maybe the Authors have a bank of samples with aliquots of plasma still available for these determinations.

Unfortunately, we do not have a bank of samples with plasma aliquots left in order to perform the suggested determinations. The samples have already been allocated to other substudies (e.g. Thorne et al., Clin Proteomics, 2020; Nielsen et al., PloS One, 2019).

Thorne AM, Huang H, Eijken M, Norregaard R, Ploeg RJ, Jespersen B, et al. Subclinical effects of remote ischaemic conditioning in human kidney transplants revealed by quantitative proteomics. Clin Proteomics. 2020;1–13. 

Nielsen MB, Krogstrup N V., Nieuwenhuijs-Moekeid GJ, Oltean M, Dor FJMF, Jespersen B, et al. P-NGAL Day 1 predicts early but not one year graft function following deceased donor kidney transplantation - The CONTEXT study. PLoS One. 2019;14(2):1–14. 

d. Organ-specific indications cannot be expected from the proposed laboratory strategy to explore plasma thiols. The investigation of individual thiol species would provide much higher chances to obtain some level of information on the transplanted organ (see later in the next point).

We agree with the reviewer that the measurement of individual thiol species would provide us with an extra level of biological information compared to the single quantification of total free thiols. However, as mentioned above, we were not able to perform these measurements. In addition, the reviewer touches upon a fair argument here that organ-specific information cannot be expected from the measurement of total free thiols. Although determination of total free thiols is rather unspecific in relation to individual oxidative stress-mediated diseases, they are very specific to systemic oxidative stress and that makes it one of the most advocated, easy and accurate ways of oxidative stress quantification in vivo. Thus, by performing this measurement, our study does show a clear association between systemic oxidative stress and early and late graft function in recipients of kidney grafts from deceased donors.

3. The changes observed in the levels of free thiols in plasma of this study and their correlation with the success of transplantation and organ function are more than expected if we consider that tubular epithelial cells are very rich in gamma-glutamyl transpeptidase or ��GT (Giustarini, Galvagni et al. 2020). Therefore, reduced or absent function of tubular epithelia cells observed in the late stages of kidney disease, is expected to impair the renal metabolism and extracellular levels of LMW thiols, and especially of Cys. Possibly, what the Authors in this study are measuring with their thiol assay in plasma is the ��GT activity of the transplanted organ that obviously is higher in successfully treated subjects.

We thank the reviewer for bringing up this interesting hypothesis. We agree that this could very well be one of the contributing mechanisms to the changes we observed in the levels of plasma free thiols and their correlations with early and one-year graft function in renal transplant recipients. Unfortunately, we don’t have gamma-glutamyl transpeptidase measured in our study, but if we would have it available, this would have been an interesting and potentially valuable secondary analysis to further complement the presented results. We have previously published results on P-NGAL, a tubular function marker, which predicted the early kidney graft function (mGFR Day 5 and the estimated time to a 50% reduction in P-creatinine) well, but failed to be able to predict graft function at three and twelve months after kidney transplantation (Nielsen et al., PlosOne, 2019).

Nielsen MB, Krogstrup N V., Nieuwenhuijs-Moekeid GJ, Oltean M, Dor FJMF, Jespersen B, et al. P-NGAL Day 1 predicts early but not one year graft function following deceased donor kidney transplantation - The CONTEXT study. PLoS One. 2019;14(2):1–14. 

4. The redox balance of extracellular thiols declines with the subject’ age (Jones, Mody et al. 2002, Giustarini, Dalle-Donne et al. 2006) and such a decline is even more rapid in case of premature aging and impaired redox homeostasis, which are characteristic conditions of CKD (Galli et al. Free Rad Res, 2014; Reggiani et al. 2015, Fanti, Giustarini et al. 2015). The results in this study (when the actual reduced form of thiols will be measured) should be corrected for the age of the patients as potential confounding factor.

We agree with the reviewer that age / aging strongly influence the levels of extracellular free thiols. In the previous studies cited in our above replies, the effect of age on thiol levels is very clear and has been replicated many times. Therefore, we agree that we should take into account the confounding effect of age. Age was therefore already included as a covariate in our linear regression models (statistics, lines 120, as well as Table 2 and 3).

5. Based on the correlation between thiols and mGFR it could be assumed that GFR could be utilized instead of thiols as a biomarker of a successful transplantation, which I guess is routine in the clinical monitoring of transplanted patients. What plasma thiols (those measured with this study) actually add up to the already available indices of organ function in transplantation protocols?

Our findings that plasma free thiols are associated with early and one-year graft function indeed raise the question whether and how this could be applied in clinical practice. The reviewer is correct that the GFR is routinely monitored in transplanted patients. However, in most cases, this will be the estimated GFR (eGFR) instead of the mGFR, which is the gold standard for measuring kidney function, but not frequently used in clinical practice as the mGFR is very time-consuming and costly to determine and is accompanied by a high patient burden as the transplant recipient needs to attend the nuclear medicine department for several hours in order to complete the determination of the mGFR. Furthermore, especially in the early days after transplantation, the kidney is not in a steady state condition and this affects both of these estimates. However, the measurement of plasma free thiols as a reflection of systemic oxidative stress may serve as an attractive screening tool as it is largely non-invasive and can be performed at relatively low costs (estimated at less than 2 dollars per assay). Nevertheless, it should be noted that further study is required to determine if plasma free thiols could be used as a real screening tool as this would depend on several circumstances, which can also change over time. For example, if a renal transplant recipient is suspected of having graft failure, a result of a relatively low concentration of plasma free thiols might be a screening result that could help to determine whether there is a necessity for further (invasive and/or costly) testing, e.g. by taking a renal biopsy.

6. The term “oxidated” should be revised and substituted with oxidized.

Thank you for this correction (line 22).

7. The Authors have disregarded most of the studies performed so far on plasma thiols in the introduction of their study and in the discussion of the results. This and other aspects discussed earlier in this revision report, demonstrate poor confidence with this topic. I suggest to refer to experts in the field of redox biology and medicine, and especially in thiol analysis, to obtain sufficed advise during the revision of their manuscript.

We regret to hear that our manuscript demonstrated poor confidence with the topic. It has never been our intention to ignore the important work of others. We have now added references on the subject to the Introduction section in the revised manuscript. Furthermore, we have had extensive discussion with redox biology experts from the University of Groningen and decided to include Arno R. Bourgonje as co-author to further improve the manuscript. Their advices are integrated into our rebuttal letter as well as the revised manuscript.

Do you want your identity to be public for this peer review? For information about this choice, including consent withdrawal, please see our Privacy Policy.

 Reviewer #1: No

Reviewer #2: Yes: Francesco Galli

---

## [Decision Letter · Decision Letter 1]

26 Mar 2021

PONE-D-20-32900R1

Elevated plasma free thiols are associated with early and one-year graft function in renal transplant recipients

PLOS ONE

Dear Dr. Nielsen,

Thank you for submitting your manuscript to PLOS ONE. After careful consideration, we feel that it has merit but does not fully meet PLOS ONE’s publication criteria as it currently stands. Therefore, we invite you to submit a revised version of the manuscript that addresses the points raised during the review process.

While reviewer #1 found your manuscript improved, reviewer #2 was not satisfied by the revision as major issues still remain unanswered. Please do revise the manuscript accordingly and provide clear an unequivocal answers to the issues raised.

We look forward to receiving your revised manuscript.

Kind regards,

Gianpaolo Reboldi, MD, MSc, PhD

Academic Editor

PLOS ONE

Reviewers' comments:

Reviewer's Responses to Questions

**Comments to the Author**

1. If the authors have adequately addressed your comments raised in a previous round of review and you feel that this manuscript is now acceptable for publication, you may indicate that here to bypass the “Comments to the Author” section, enter your conflict of interest statement in the “Confidential to Editor” section, and submit your "Accept" recommendation.

Reviewer #1: All comments have been addressed

Reviewer #2: All comments have been addressed

2. Is the manuscript technically sound, and do the data support the conclusions?

Reviewer #1: (No Response)

Reviewer #2: Partly

3. Has the statistical analysis been performed appropriately and rigorously? 

Reviewer #1: (No Response)

Reviewer #2: I Don't Know

4. Have the authors made all data underlying the findings in their manuscript fully available?

Reviewer #1: (No Response)

Reviewer #2: Yes

5. Is the manuscript presented in an intelligible fashion and written in standard English?

Reviewer #1: (No Response)

Reviewer #2: Yes

6. Review Comments to the Author

Reviewer #1: (No Response)

Reviewer #2: The Authors have not addressed the concerns of this Reviewer. The main limits highlighted during the previous round of revision remain there.

7. PLOS authors have the option to publish the peer review history of their article (what does this mean?). If published, this will include your full peer review and any attached files.

Reviewer #1: No

Reviewer #2: No

---

## [Author Response · Author response to Decision Letter 1]

20 Apr 2021

Dear reviewer #2

First of all, we would to express our gratitude again for your previous comments and suggestions on our manuscript, which were highly appreciated by all authors. We regret to hear that you were not fully satisfied with our revised manuscript. As we considered your suggestions and critiques to be very relevant, we have done our utmost best to address them carefully in order to optimise our manuscript and submit a successful revision. However, you have outlined several concerns about our study, mainly from a redox biological perspective, which we have all extensively addressed in our first rebuttal letter. Unfortunately, however, we were not able to completely resolve all of them, mainly due to the lack of available resources (i.e. sample volumes for measuring additional thiol compounds or laboratory indices). For the current revision, we were a bit unsure which main limits you are precisely referring to in your above statement, but we assume that they relate to the following two issues:

1. Lack of additional experiments and laboratory indices: 

We agree with the reviewer that from a redox biology perspective, it would be highly interesting and relevant to perform more granular analyses by focusing on individual thiol species and include extra indices, which may be differentially affected by a certain clinical event and provide a deeper understanding of redox dynamics. Although we would have strongly considered to perform these experiments when we had sufficient biomaterials left, we are forced to accept this as a limitation of our study. Unfortunately, we do not have a bank of samples with aliquots left as remaining ones have already been allocated to other studies. This led us to decide to devote a substantial part of the Discussion section to your suggestions, so we could provide that as an inspiration for future studies. For further details see our previous replies to comments 2a, 2b, 2c, and 2d.

2. The measurement method of free thiols:

In our present study, we have determined total free thiol content in plasma by derivatization with DTNB as thiol-reactive agent. This is based on the standardised Ellman reaction. Using this method, we measure the combination of protein-bound free thiols (since proteins were not removed from the samples) and low-molecular-weight (LMW) free thiols, e.g. cysteine, glutathione and homocysteine. You are fully correct that total thiol content (including oxidized thiols with disulphide bridges) was not measured in our study. In that case, we should have used a stronger reducing agent like dithiothreitol (DTT), which is able to additionally reduce disulphide bonds and oxidized protein-bound thiols(1).However, we can reassure you that our measurement method has been well-developed, optimized and validated in our center and in literature. Over the past couple of years, extensive quality control experiments have been performed to ensure that samples are in optimal conditions and the generated results are trustworthy, e.g. the method is accompanied by high stability as repeated freeze-thaw cycles confer no major effects on free thiol levels of individual samples. Furthermore, our method has proven its power to predict clinical parameters in many previous studies(2–6), in which the levels of free thiols were in accordance with the concentration ranges as observed within the present study, using the exact same detection method. Finally, our method has been employed in different centers (Switzerland e.g.), while measurement results remain consistent, ruling out a potential location/environment-specific variation as well. Usually, mean total free thiols in serum/plasma add up to 400-600 μM(7), but this may vary by disease condition, ranging from 100 to 1000 μM (as can be detected by the L-cysteine calibration curve, see Materials & Methods).

Although you are absolutely correct that some major limitations will remain there, we have made every possible effort and used all available resources to achieve the best outcome for our study. Accordingly, we have improved the manuscript as carefully as possible based on your comments and suggestions. Additionally, as you suggested, we included additional expertise in redox biology and medicine, which also led to an updated list of authors. Again, we regret to be informed that you were not fully satisfied with our rebuttal. However, in light of all these considerations, we sincerely hope for your understanding and that you will re-consider your evaluation of our manuscript in light of PLoS ONE’s publication criteria.

Thank you for taking the time and effort to evaluate our manuscript. We look forward to receive your response.

1. Sutton TR, Minnion M, Barbarino F, Koster G, Fernandez BO, Cumpstey AF, et al. A robust and versatile mass spectrometry platform for comprehensive assessment of the thiol redox metabolome. Redox Biol. 2018 Jun 1;16:359–80. 

2. Abdulle AE, Bourgonje AR, Kieneker LM, Koning AM, La Bastide-Van Gemert S, Bulthuis MLC, et al. Serum free thiols predict cardiovascular events and all-cause mortality in the general population: A prospective cohort study. BMC Med. 2020 May 27;18(1). 

3. Bourgonje AR, von Martels JZH, Bulthuis MLC, van Londen M, Faber KN, Dijkstra G, et al. Crohn’s Disease in Clinical Remission Is Marked by Systemic Oxidative Stress. Front Physiol. 2019 Apr 26;10:499. 

4. Koning AM, Meijers WC, Pasch A, Leuvenink HGD, Frenay A-RS, Dekker MM, et al. Serum free thiols in chronic heart failure. Pharmacol Res. 2016 Sep;111:452–8. 

5. Frenay A-RS, de Borst MH, Bachtler M, Tschopp N, Keyzer CA, van den Berg E, et al. Serum free sulfhydryl status is associated with patient and graft survival in renal transplant recipients. Free Radic Biol Med. 2016;99:345–51. 

6. Damba T, Bourgonje AR, Abdulle AE, Pasch A, Sydor S, van den Berg EH, et al. Oxidative stress is associated with suspected non-alcoholic fatty liver disease and all-cause mortality in the general population. Liver Int. 2020 Sep 1;40(9):2148–59. 

7. Turell L, Radi R, Alvarez B. The thiol pool in human plasma: the central contribution of albumin to redox processes. Free Radic Biol Med. 2013 Dec;65:244–53.

---

## [Decision Letter · Decision Letter 2]

28 Jul 2021

Elevated plasma free thiols are associated with early and one-year graft function in renal transplant recipients

PONE-D-20-32900R2

Dear Dr. Nielsen,

We’re pleased to inform you that your manuscript has been judged scientifically suitable for publication and will be formally accepted for publication once it meets all outstanding technical requirements.

Kind regards,

Gianpaolo Reboldi, MD, MSc, PhD

Academic Editor

PLOS ONE

Additional Editor Comments (optional):

Reviewers' comments:

Reviewer's Responses to Questions

**Comments to the Author**

1. If the authors have adequately addressed your comments raised in a previous round of review and you feel that this manuscript is now acceptable for publication, you may indicate that here to bypass the “Comments to the Author” section, enter your conflict of interest statement in the “Confidential to Editor” section, and submit your "Accept" recommendation.

Reviewer #2: All comments have been addressed

Reviewer #3: All comments have been addressed

2. Is the manuscript technically sound, and do the data support the conclusions?

Reviewer #2: No

Reviewer #3: Yes

3. Has the statistical analysis been performed appropriately and rigorously? 

Reviewer #2: N/A

Reviewer #3: Yes

4. Have the authors made all data underlying the findings in their manuscript fully available?

Reviewer #2: Yes

Reviewer #3: Yes

5. Is the manuscript presented in an intelligible fashion and written in standard English?

Reviewer #2: Yes

Reviewer #3: Yes

6. Review Comments to the Author

Reviewer #2: it is my opinion that the limits of this study, addressed since the previous round of revision, do not allow the publication in this brad-spectrum Journal as regular original article. By admission of the Authors, these limits derive from the available resources and competencies that are obviously insufficient to reach original and relevant data. Also, the association between thiols and GFR is not such surprising and however the prediction power of measuring thiols on the graft function has not sufficiently been demonstrated being the study design and statistic power other limits that have not been addressed.

May be the Authors could convert this study in a different format; for example a letter to the Editor in a nephrology journal could be an option.

Reviewer #3: The subject of this study is interesting and in line with currently literature. In general, this study is well conducted, and the paper is very well written.

7. PLOS authors have the option to publish the peer review history of their article (what does this mean?). If published, this will include your full peer review and any attached files.

Reviewer #2: No

Reviewer #3: No

---

## [Editor Report · Acceptance letter]

2 Aug 2021

PONE-D-20-32900R2 

Elevated plasma free thiols are associated with early and one-year graft function in renal transplant recipients 

Dear Dr. Nielsen:

I'm pleased to inform you that your manuscript has been deemed suitable for publication in PLOS ONE. Congratulations! Your manuscript is now with our production department. 

Kind regards, 

on behalf of

Prof Gianpaolo Reboldi 

Academic Editor

PLOS ONE